# TVAE: Triplet-Based Variational Autoencoder using Metric Learning

**Haque Ishfaq** [*]
Department of Statistics
Stanford University
`hmishfaq@stanford.edu`

**Assaf Hoogi**[*]**& Daniel Rubin**
Department of Radiology
Stanford University
`{ahoogi,dlrubin}@stanford.edu`

## Abstract

Deep metric learning has been demonstrated to be highly effective in learning semantic representation and encoding information that can be used to measure data similarity, by relying on the embedding learned from metric learning. At the same time, variational autoencoder (VAE) has widely been used to approximate inference and proved to have a good performance for directed probabilistic models. However, for traditional VAE, the data label or feature information are intractable. Similarly, traditional representation learning approaches fail to represent many salient aspects of the data. In this project, we propose a novel integrated framework to learn latent embedding in VAE by incorporating deep metric learning. The features are learned by optimizing a triplet loss on the mean vectors of VAE in conjunction with standard evidence lower bound (ELBO) of VAE. This approach, which we call Triplet based Variational Autoencoder (TVAE), allows us to capture more fine-grained information in the latent embedding. Our model is tested on MNIST data set and achieves a high triplet accuracy of 95.60% while the traditional VAE (Kingma & Welling, 2013) achieves triplet accuracy of 75.08%.

## 1 Introduction

Learning semantic similarity between pairs of images is a core part of visual competence and learning. When applied on proper embedding of input data, similarity metric functions such as Euclidean distances, Mahalanobis distance, cosine similarity etc result in superior metric for similarity measure and reduce many complex classification problems to simple nearest neighbor problems. But these same similarity metric functions would perform poorly when applied on raw complex input datasets. Image embeddings learned as a part of larger classification task using deep nets have various practical limitations for several scenarios. In extreme classification problems (Choromanska et al., 2013; Bengio et al., 2010) where the number of possible categories is very large or possibly unknown, conventional classification learning approaches are essentially useless since the availability of training examples for each class becomes scarce, if not totally unavailable. Hence, a new line of approach, namely metric learning (Schroff et al., 2015; Oh Song et al., 2016; Huang & Peng, 2017) has gained much popularity for its ability to learn image embedding directly using the concept of relative distances rather than relying on specific category information. This way, it is able to learn a metric space where nearest neighbor based methods would naturally give superior performance due to the higher quality representation of input images in the learned embedding space. This approach has the potential to improve the way generative models such as Variational Autoencders (Kingma & Welling, 2013; Rezende et al., 2014) are learned. While VAE can perform extremely efficient approximate inference in latent Gaussian model, the latent embedding space it learns lacks many salient aspects of the original data. Motivated from Triplet Network as explained in Hoffer & Ailon (2015), in this project, we propose a new architecture and a loss function for training VAE, which is capable of two tasks at the same time - learning latent image representations with fine-grained information and doing stochastic inference.

---

[*]These two authors contributed equally

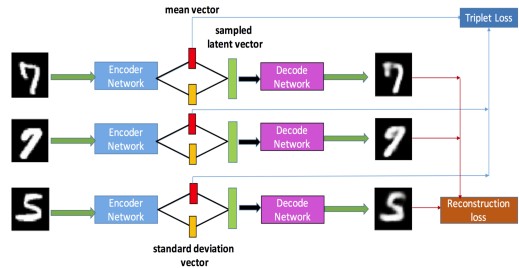

Figure 1: Model overview. As input a triplet of digit images (7,7,5) is given to three identical encoder networks. The mean latent vectors of three input images are used to calculate the triplet loss and the reconstructed images by the identical decoders are used to calculate the reconstruction error.

## 2 TRIPLET-BASED VARIATIONAL AUTOENCODER

Our proposed hybrid model in Fig.1 is motivated as a way to improve VAE, so that it can learn latent representation enriched with more fine-grained information. To achieve this we optimize the network by minimizing the upper-bound on the expected negative log-likelihood of data and triplet loss simultaneously.

The encoder in VAE encodes an image $x$ to a latent vector $z = Encoder(x) \sim q(z|x)$. The decoder decodes the latent vector $z$ back to an image $\bar{x} = Decoder(z) \sim p(x|z)$. To regularize the encoder, the VAE imposes a prior over the latent distribution $p(z)$. The VAE loss consists of two parts: the reconstruction loss and the KL Divergence loss. The reconstruction loss $\mathcal{L}_{rec} = -\mathbb{E}_{q(z|x)}[\log p(x|z)]$ is the negative expected log-likelihood of the observations in x. And the KL-Divergence loss $\mathcal{L}_{KL} = \mathrm{KL}[q(z|x)||p(z)]$ characterizes the distance between the distribution $q(z|x)$ and prior distribution.

In each iteration of training, the input triplet $(x_a, x_p, x_n)$ is randomly sampled from the training set in such a way that the anchor $x$ is more similar to the positive $x_p$ than the negative $x_n$. Then the triplet of three images are fed into encoder network simultaneously to get their mean latent embedding $f(x_a)$, $f(x_p)$ and $f(x_n)$. We then define a loss function $\mathcal{L}_{triplet}(\cdot)$ over triplets to model the similarity structure over the images as in Wang et al. (2014). The triplet loss can be expressed as

$$\mathcal{L}_{triplet}(x_a, x_p, x_n) = \max\{0, D(x_a, x_p) - D(x_a, x_n) + m\}, \tag{1}$$

where $D(x_i, x_j) = ||f(x_i) - f(x_j)||_2$ is the Euclidean distance between the mean latent vector of images $x_i$ and $x_j$ and m is threshold margin. Thus our final loss function for an input triplet is given by:

$$\mathcal{L}_{TVAE} = \mathcal{L}_{rec} + \mathcal{L}_{KL} + \mathcal{L}_{triplet} \tag{2}$$

## 3 EXPERIMENTS

We focus our experiments on preservation of the semantic structure in the learned latent embedding and image generation ability compared to original VAE in Kingma & Welling (2013). For experiments on MNIST (LeCun et al., 1998), we adopted a simple network structure with two fully connected layers as encoder and decoder and used pixel-to-pixel $L2$ distance loss function as reconstruction loss. The dimension of the latent embedding space was 20.

Table 1: Triplet accuracy on MNIST

| Model | Triplet Accuracy |
|---|---|
| VAE (Kingma & Welling, 2013) | 75.08% |
| Triplet VAE | 95.60% |

## 4 RESULTS

We visually explore the learned embedding distribution for the mean vector. With an additional triplet loss term, the mean vectors from different groups are more compactly clustered, as shown in Fig. 2b. On the other hand, without the added triplet loss, the image clusters are less compact and seem to be spreading out in the spatial space as seen in Fig. 2a. In this case, we also observe that images from one class are more likely to be divided into multiple small clusters and images from different clusters overlaps with each other more often.

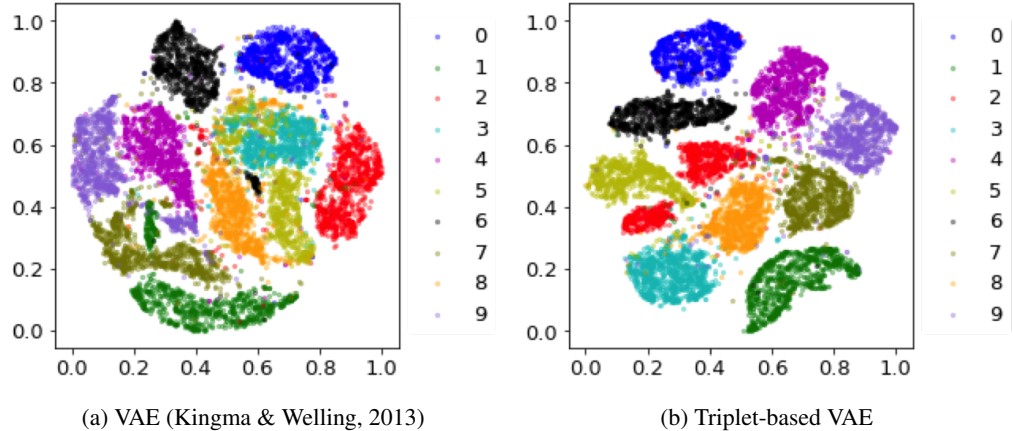

(a) VAE (Kingma & Welling, 2013)          (b) Triplet-based VAE

Figure 2: t-SNE projection of the latent mean vector for MNIST test dataset.

In order to evaluate the structure quality in terms of preserved relative distance among different classes, we analyze learned latent embedding of unseen triplets. In Table 1 we calculate triplet accuracy which is defined by the percentage of triplets that incur a loss of zero in Eq.1. We see that using TVAE, for 95.60% of test triplets, we get learned latent embedding which maintain the relative distances among classes. On the other hand, for traditional VAE, we preserve this relative distances for only 75.08% of test triplets.

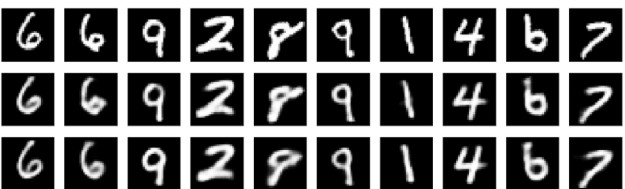

Figure 3: Comparison of reconstructed images from the MNIST dataset. The first row is the input images from the MNIST test set. The second row is the reconstructed images generated by the plain VAE. The third row is the reconstructed images generated by the TVAE.

## 5 DISCUSSION

Triplet based Variational Autoencoders (TVAEs) provide a new set of tools for learning latent embedding and performing approximate inference that leverage both traditional VAE and deep metric learning techniques. By incorporating triplet constraint in the learning process, TVAEs can learn an interpretable latent representation that preserves semantic structure of the original dataset. Our method provides an initial framework for learning latent embedding that would be able to encode various notions of similarity. We demonstrate that TVAE generates high quality samples as good as the traditional VAE while encoding more semantic structural information in the latent embedding. Our future work will include analysis of medical datasets.

ACKNOWLEDGMENTS

This work was supported in part by grants from the National Cancer Institute, National Institutes of Health, 1U01CA190214 and 1U01CA187947.

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
