# OpenReview forum: "TVAE: Triplet-Based Variational Autoencoder using Metric Learning"
_ICLR.cc/2018/Workshop — Reject_

### Official Review · AnonReviewer3 · 2018-03-10
**A possibly interesting addition to VAE. Experiments are insufficient.**

**Rating:** 3
**Confidence:** 5

**Review:**

The paper proposes adding a supervised signal to VAE signal which is meant to improve the usability of the latent space vectors.  Specifically, the proposal is a triplet-based loss first used by Frome et al. (2007), which has since been widely used in metric learning (example Weinberger & Saul 2009). This loss relies on a weak supervision telling us "A is more similar to B than it is to C". The strength of this loss function is in cases where direct labels are unavailable, but in can be used in fully labeled datasets as well, as done in this paper. The paper would be improved by finding cases where direct supervision is not available but relative comparisons are, e.g. ranking- or preference-feedback tasks.

Pros:
1. Possibly interesting idea for making the latent space vectors of a VAE more useful.

Cons:
1. Supervised or semi-supervised VAEs could just as well create better representations of the classes. There is no comparison with them or in fact no mention of this possibility.
2. Tying in with the previous point. the TVAE has strictly more information than VAE, since it has access to label information through the triplet loss, making comparisons less meaningful.
3. The triplet loss in itself is not interesting at all. I would like to see experiments comparing performance of tasks for which a good metric on the latent space is useful. For example, NN classification using latent vectors, one-shot learning of new classes, or interpretability tasks.

---

### Official Review · AnonReviewer1 · 2018-03-12
**VAE + triplet margin loss**

**Rating:** 5
**Confidence:** 5

**Review:**

This paper proposes a deep embedding approach by minimizing the
combination of the loss of VAE and triplet-based large-martin hinge
loss. Experimental results show that, based on triplet loss, the combined
approach performs much better than VAE.

However, minimizing triple-based large-margin loss on top of a deep neural
network (a deep autoencoder pertained with RBMs) for metric learning and
data visualization was proposed in a paper entitle A Deep Non-Linear
Feature Mapping for Large-Margin kNN Classification about ten years ago,
which was re-implemented using a modern Deep CNN architecture for image
ranking (Wang et al., 2014) and metric learning (Hoffer and Ailon, 2015).
The contribution in this paper is very limited, although VAE is different
from traditional autoencoder.

Moreover, it is not surprising that the triplet-based methods in this paper and in
all the three previous related papers perform much better than their respective
unsupervised counterparts, because label information is employed during the
process of triplet construction. It is better to elaborate how the supervised label
information is used to improve the experiment part.  And it is better to show the
standard kNN classification error rate on the test dataset besides the triplet-based
error rate. The experimental results on MNIST don’t seem to be convincing compared
to previous metric learning results.

In summary, the proposed method is a simple combination of existing work and
lacks novelty and insight.

---

### Official Review · AnonReviewer2 · 2018-03-13
**Paper presents an interesting model for weakly supervised learning which has been propsoed before.**

**Rating:** 2
**Confidence:** 5

**Review:**

The authors propose to combine triplet learning with variational autoencoders in a reasonable framework and demonstrate it at MNIS toy examples.

Unfortunately, this work is subsumed by the ICLR 2016 paper 'Bayesian Representation Learning With Oracle Constraints' by Karaletsos et al which has proposed the combination of tripelts with a VAE in a principled joint modeling framework.

In addition, the ICLR 16 paper proposed mechanisms to address shortcomings of the combination of a VAE with triplets related to disentangling, which this workshop paper fails to recognize.

As such, this paper unfortunately suffers from a lack of novelty.

---

### Decision · Program_Chairs · 2018-03-20
**ICLR 2018 Workshop Acceptance Decision**

**Decision:**

Reject

**Comment:**

Based on the reviews, this paper has not been accepted for presentation at the ICLR workshop. However, the conversation and updates can continue to appear here on OpenReview.